# Multi-Exoskeleton Performance Evaluation: Integrated Muscle Energy Indices to Determine the Quality and Quantity of Assistance

**DOI:** 10.3390/bioengineering11121231

**Published:** 2024-12-05

**Authors:** Vasco Fanti, Sergio Leggieri, Tommaso Poliero, Matteo Sposito, Darwin G. Caldwell, Christian Di Natali

**Affiliations:** 1Department of Advanced Robotics (ADVR), Istituto Italiano di Tecnologia (IIT), 16163 Genova, Italy; sergio.leggieri@iit.it (S.L.); tommaso.poliero@iit.it (T.P.); matteo.sposito@iit.it (M.S.); darwin.cadwell@iit.it (D.G.C.); christian.dinatali@iit.it (C.D.N.); 2Department of Informatics, Bioengineering, Robotics and Systems Engineering (DIBRIS), Università degli Studi di Genova (UniGe), 16145 Genova, Italy

**Keywords:** integrated EMG, muscular analysis, occupational exoskeletons, performance evaluation, physical human–robot interaction

## Abstract

The assessment of realistic work tasks is a critical aspect of introducing exoskeletons to work environments. However, as the experimental task’s complexity increases, the analysis of muscle activity becomes increasingly challenging. Thus, it is essential to use metrics that adequately represent the physical human–exoskeleton interaction (pHEI). Muscle activity analysis is usually reduced to a comparison of point values (average or maximum muscle contraction), neglecting the signals’ trend. Metrics based on single values, however, lack information about the dynamism of the task and its duration. Their meaning can be uncertain, especially when analyzing complex movements or temporally extended activities, and it is reduced to an overall assessment of the interaction on the whole task. This work proposes a method based on integrated EMGs (iEMGs) to evaluate the pHEI by considering task dynamism, temporal duration, and the neural energy associated with muscle activity. The resulting signal highlights the task phases in which the exoskeleton reduces or increases the effort required to accomplish the task, allowing the calculation of specific indices that quantify the energy exchange in terms of assistance (AII), resistance (RII), and overall interaction (OII). The method provides an analysis tool that enables developers and controller designers to receive insights into the exoskeleton performances and the quality of the user-robot interaction. The application of this method is provided for passive and two active back support exoskeletons: the Laevo, XoTrunk, and StreamEXO.

## 1. Introduction

The growth of interest in wearable robotics, particularly in exoskeletons, is now a proven reality [1]. Biomechanically robotic exoskeletons are structures that interact with the user’s musculoskeletal system to mutually transfer energy and power [2,3,4]. In industry, exoskeletons for back assistance aim to support the back during manual material handling activities [5,6] and prevent musculoskeletal disorders in the lumbar and sacral regions [7]. To evaluate their effectiveness, testing in real-world tasks is crucial [8]. Indeed, it is necessary to focus the assessment on realistic work tasks [9]. A proper selection of these tasks will enable the reproduction of characteristic movements associated with the work activity, as well as all these secondary movements that occur in a relevant environment (e.g., unforeseen contingencies and obstacles to be avoided). This approach facilitates a faithful assessment of the real working task in a still controllable laboratory context.

The core element for assessing the exoskeleton performance is the evaluation of the physical human–exoskeleton interaction (pHEI) [10]. This interaction consists of the exchange of energy, force, and torque. Therefore, it is vital to evaluate if such interactions occur optimally (i.e., if the exoskeleton delivers the proper amount of torque in the right moments) or not. With a suitable control strategy, the assistance provided by the exoskeleton will reduce the user’s muscle activity and joint torque. Conversely, with a poor choice of timing or torques, the exoskeleton will impede the user’s movements, increasing muscle activity and user-generated joint torque [6]. The study of joint torques and internal forces can certainly provide useful information about the pHEI [11]. However, these studies rely on kinetic and musculoskeletal models that are limited by approximations due to the complexity of the systems they represent [12]. For this reason, monitoring muscle activities remains the simplest and most reliable direct measure for assessing the energy exchanges that occur between the exoskeleton and the user [13,14,15]. Still, since electromyography could be affected by poor repeatability, special care must be taken when collecting and processing data by carefully following the standards for electromyography [16] and collecting the data on a statistically significant pool of subjects.

The evaluation metrics associated with muscle activity assessment are usually the maximum peak, average, or median of the muscle activation [13]. These metrics, however, may not fully represent the task and quantify the pHEI because they consider neither the duration of the task nor the profile of the muscle activity but just its distribution. To increase the amount of information provided by the analysis of muscle activity, we believe that metrics based on signal distribution should be supported by new indices and visualization methods that assess the change in muscle activity at each stage of the task while taking into account the duration and energy expended during the interaction.

To perform a comprehensive analysis that includes the development of indices that consider both the activity’s dynamism and its temporal extent, we propose a data analysis technique based on integrated EMG (iEMG) [17]. With this process, the duration of the signal is considered, and muscle activity reduction is weighted based on the integral of the EMG signal. As a result, the final signal is proportional to the duration of the activity under analysis, the muscles’ exertion, and thus, the total neural energy contribution [18] required by the motor neuron to activate the muscle fibers during the task. The resultant processed signal can be used to evaluate pHEI in terms of both assistance and resistance to the movement [11] at each phase of the task. Moreover, assistive and resistive contributions are quantified with specific indices that can be calculated by considering the task’s total duration or any part of it.

The proposed methodology consists of a set of equations to extract hidden information concerning the energy exchange that occurs between the user and the exoskeleton. The methodology is presented and validated by comparing it with classical indices. The tool provides essential insights about the pHEI while quantifying the total muscle assistance/resistance elicited by the interaction with the exoskeleton during each phase of the task. After validation, the analysis is applied to the data collected in the laboratory from real workers during the testing of three back-support exoskeletons: Laevo, XoTrunk, and StreamEXO.

## 2. Methods

To evaluate the reduction in muscle activity, the EMG signal acquired while wearing the exoskeleton (Exo) is usually compared with the baseline condition, i.e., the EMG signal recorded without wearing the exoskeleton (NoExo).
(1)$$PV=\frac{VAL_{NoExo}-VAL_{Exo}}{VAL_{NoExo}}*100.$$

A common approach for quantifying muscle activity reduction is calculating the percentage variation (PV) [19], which is an index based on the characteristic values of the data distribution such as mean, median, 90th percentile, etc. With this approach, a single value representative of the whole task is extracted from a test of the user wearing an exoskeleton in a controlled environment [13]. Once the characteristic value (VAL) is selected, the PV can be calculated by computing the difference between $VAL_{NoExo}$ and $VAL_{Exo}$. The result is then normalized relative to $VAL_{NoExo}$, as shown in (Equation 1).

It is worth underlining that this method requires knowledge of the probability distribution of the data (e.g., normal, binomial, Poisson, etc.) to understand the meaning of the VAL metric considered. Therefore, evaluating the distribution of the signals of interest is critical. However, if the task involves complex movements, time-extended tasks, or execution in uncontrolled environments, a complex EMG signal with an irregular distribution is expected. This aspect will be addressed in more detail in Section 4.

### IBA: Integral-Based Assistance

The IBA method relies on indices calculated on the full NoExo and Exo signals to evaluate the exoskeleton’s impact on the user musculature during the entire task duration, as shown in Figure 1a.
(2)$$I\text{\_}Trend\left(i\right)=\frac{EMG_{NoExo}\left(i\right)*\delta_{t}-EMG_{Exo}\left(i\right)*\delta_{t}}{\left[\sum_{i=1}^{N}EMG_{NoExo}\left(i\right)\right]*\Delta_{t}}*100.$$

Thus, the numerator signals in (Equation 2) are considered piecewise continuous, and the indices are calculated afterward. The denominator reports the NoExo EMG’s integral [18], so it normalizes the numerator to the neural energy expenditure associated with the task (i.e., the energy required by the motor neuron to contract the muscle fibers). Therefore, if the iEMG normalizes the difference $EMG_{NoExo}-EMG_{Exo}$ (the EMG pre-processing is reported in Section 3.3), the resultant is proportional to the activity duration and the overall neural energy the muscles require to perform the task. Since the operations are performed on full signals and not on values, the result of the equation’s integral-based trend (I_Trend) is still a signal. For this reason, even small desynchronizations between Exo and NoExo signals can lead to very different I_Trend values. Thus, to apply this method, it is critical to ensure that the signals are synchronous with each other in all phases of the tasks and sub-tasks. If necessary, after the data segmentation, it is suggested that the number of samples that make up the signals be scaled. The I_Trend calculation is reported in (Equation 2) where $\delta_{t}$ represents the sampling time, $\Delta_{t}$ is the duration of the task, *i* is the number of samples in each signal, and *N* is their total number.

The effectiveness of (Equation 2) is provided by the normalized numerator, which allows the contribution of the exoskeleton to be evaluated at each stage of the task, as visible in Figure 1b–e. The I_Trend, in fact, will present positive values when there is a net reduction in the muscular activation of the user. We define this as an assistive period during the task [11]. On the other hand, values of the I_trend are negative if the exoskeleton provides torque that is in opposition to the user’s movement, i.e., providing resistance. These instantaneous contributions can be collected under performance indices that globally evaluate the activity in terms of assistance, resistance, and overall interaction. The assistive interaction index (AII) quantifies the muscle assistance provided by the exoskeleton by measuring the reduction in muscle activity, and it is calculated by summing all the positive values of the I_Trend, as shown in (Equation 3). The resistive interaction index (RII) analyzes the increase in muscle activity by quantifying the resistance to the movement. It is calculated as the summation of the absolute value of the negative values of the I_Trend (Equation 4). Finally, the overall interaction index (OII) is useful for evaluating the balance between assistance and resistance and can be calculated by subtracting the RII from the AII (Equation 5).
(3)$$AII=\sum_{i}I\text{\_}Trend(i)\;\mathrm{if}\;I\text{\_}Trend(i)\geq 0,$$
(4)$$RII=\;|\sum_{i}I\text{\_}Trend(i)|\;\mathrm{if}\;I\text{\_}Trend(i)<0,$$
(5)$$OII=AII-RII=\sum_{i}I\text{\_}Trend(i).$$

The methodology can be applied to the full signal to provide representative indices for a whole activity, as shown in Figure 1b, or to select portions to investigate the exoskeleton performance during characteristic tasks, as shown in Figure 1c–e. Moreover, it can be applied to any body district that experiences changes in muscle activations induced by interaction with an exoskeleton.

Summarizing the above, with this method, we expect to do the following: (i) separately evaluate and quantify the assistive and resistive contributions given by the interaction with the exoskeleton; (ii) monitor at each instant of the task the effect of the exoskeleton on the user; (iii) verify the correspondence between data/results with the physical reality observed during task execution; (iv) consider the time duration of the test and thus the effectiveness of the exoskeleton over time; (v) quantify the benefit of the physical human-exoskeleton interaction based on the reduction of muscle energy required to activate the muscles under investigation.

## 3. Experimental Setup

### 3.1. Exoskeletons

Three different back support exoskeletons, Laevo, Xotrunk, and StreamEXO, were used for this work.

Laevo, Figure 2a, weighs 3 kg. This is a rigid, passively actuated, commercial exoskeleton that can be adjusted in height and width through straps. As it is passive and relies on gas springs, its control strategy is limited and proportional to the flexion angle of the back [20]. To ensure freedom of movement, the control strategy was set to provide assistance after exceeding 15 degrees of trunk bending angle.

Xotrunk, Figure 2b, weighs 8.2 kg. It is also rigid and can be adjusted in height, but the hip width is fixed. It is actively actuated using two brushless DC motors (Maxon EC 60 flat) powered by a 12-cell, 48-volt battery. The combined motors can deliver torque up to 40 Nm. Although capable of following software-modulated control, the XoTrunk during this work uses a control strategy that combines the contribution of the bending angle and the forearm muscle activity. The forearm is monitored with a MYO armband [21] that can detect muscle activation and subsequently provide a torque boost to aid the trunk extension. Moreover, once the user stands upright (i.e., if the trunk flexion angle is below 15 degrees), the exoskeleton does not provide any torque, and the user can move the legs freely without any hindrance [20].

StreamEXO was designed in STREAM (https://streams2r.eu/) (accessed on 4 January 2022) and further improved in BEEYONDERS (https://beeyonders.eu/) (accessed on 24 April 2024), as shown in Figure 2c.It weighs 7.5 kg, and its design features a one-size-fits-all solution to ensure maximum fit and comfort from the 5th to 95th percentiles of the European workers’ population. With this solution, although the exoskeleton is rigid, the hips’ width and the exoskeleton’s full height can be easily adjusted without using straps. StreamEXO is a specific exoskeleton solution tailored to railway and construction workers’ needs. Therefore, design solutions were optimized to ensure the maximum range of fit and lowest bulk. This optimization was also implemented in the software, allowing customized control strategies. These control strategies were specifically developed [22] and optimized for human-in-the-loop [23] operations to address the activities performed by railway and construction workers. They were also designed to differentiate between dynamic and static activities.

### 3.2. Work Tasks

Two work activities were analyzed in this experimental session; one was dynamic (“Gross” positioning), and the other was static (“Fine” positioning) [22,24].

The “Gross” positioning work task is shown in Figure 3a. It consists of lifting a cable duct (whose characteristics are given in Section 3.3) from the ground level and carrying it at waist height with two hands for approximately 2 m before lowering it again to the ground, as shown in Figure 3b,c. The worker repeats this lifting, carrying, and lowering routine 10 times.

The “Fine” positioning of the cable duct is shown in Figure 3d. During this phase, the worker positions himself with his legs spread and his back bent, as shown in Figure 3e,f. The worker lifts the cable duct a few centimeters, moving it from close to the left leg to the right leg. Motion is primarily lateral, although some forward/backward motion may be needed for accurate positioning/alignment. The worker remains bent over for approximately 8 s before standing up. This motion is repeated 10 times, alternately between moving the cable duct from close to the left to the right leg and vice versa at each repetition. This replicates the typical worker motion while placing the cable ducts precisely within the trackside trench. Although the downward and upward bending phases are dynamic, the task is considered static because the longest duration, and the aspect of key interest for our analysis is when the worker maintains the forward flexed position. In real work scenarios, workers prefer to settle several ducts consecutively while remaining bent over instead of standing up and down at each block.

### 3.3. Participants, Sensors, and Data Processing

Laboratory testing was performed to evaluate the exoskeletons’ efficacy through scientific analysis in a controlled environment. The test was performed in accordance with the experimental protocol approved by the Ethics Committee of Liguria, Italy (protocol number: 001/2019), and complied with the Helsinki Declaration. A pool of volunteers was selected by a partner company, MerMec STE, as is characteristic of the industrial sector. We tested 10 male professional railway workers with the following characteristics: age 45 ± 10 years, height 177 ± 7 cm, and weight 84 ± 14 kg. The cable ducts measured 50 cm in length, 20 cm in width, and 14 cm in thickness. Their weight was 19 kg. The testing used IMUs-based motion capture sensors (Xsens Awinda 3D Wireless Motion Tracker, Xsens Technologies B.V., Enschede, The Netherlands) to collect data from the feet, lower legs, upper legs, pelvis, and sternum and to segment the signals. This was synchronized with the measurement of muscle activation, recorded with Wi-Fi superficial EMG sensors (FreeEMG300 System, BTS, Milan, Italy). We monitored the muscle activity in the Erector Spinea Ileocostalis, Erector Spinae Longissimus, Biceps Femoris, and Recutus Femoris, both on the right and left side, following the SENIAM guidelines [16]. The analyzed muscle selection is related to the direction assisted by the exoskeletons and the body sections where they act. The collected signals were processed using MATLAB software (MATLAB 9.7.0, MathWorks, Natick, MA, USA). Muscle signals were filtered using a fourth-order Butterworth filter with a bandwidth between 30 Hz and 400 Hz. These were rectified and filtered again with a Butterworth fourth-order low-pass filter with a cutoff frequency of 2.5 Hz [25]. Before testing, the maximum voluntary contraction (MVC) of each muscle was recorded. The 95th percentile was calculated and used to normalize the EMG signals. Signal synchronization was ensured by recording the data from different equipment by connecting them with a trigger system that enables the equipment to start recording at the same temporal instant. Kinematic and EMG signals were segmented and thus aligned by using thresholds on the hip and sternum joint rotations. Since the static task was timed, subtask synchronization was ensured accordingly. When this is not the case, as in the dynamic task, sub-segmentation based on joint rotation is suggested to identify the task phases common to both signals. Then, the signals were resampled to ensure that movements at different speeds/dynamics had the same number of samples and, thus, were comparable in each phase of the task.

## 4. Metrics Validation

To show the effectiveness of the methods presented in this paper, we propose some comparisons with standard index-based approaches calculated on the data from a single subject shown in Figure 1a. First, we evaluate the EMG signals by calculating the muscle activity PV (percentage variation) using the values of average (Mean) and maximum (Max) muscle activations. Then, we report the three indices calculated using the IBA method on the full task and its sub-segmentations, as shown in Figure 1b–e. The analysis ends by comparing standard indices with the proposed IBA indices.

### 4.1. Results

#### 4.1.1. Average and Maximum Muscle Activity Reduction

Considering the EMG signals, Figure 1a reports the muscle activity corresponding to the full task execution, and the black vertical lines correspond to the segmentation instants that define the bending, position holding, and rising up phases, respectively. In the full task, as reported in Table 1, the average muscle activation (Mean) for the NoExo signal is MeanNoExo = 26.63 [%MVC], while for the Exo signal is MeanExo = 20.98 [%MVC]. The resulting PV is MeanPV = 21.23 [%]. The same analysis was conducted considering the sub-tasks of bending, position holding, and rising up, as shown in Table 1. During bending MeanPV = 29.12 [%], during position holding MeanPV = 18.49 [%], and finally, during rising up MeanPV = 10.33 [%]. Similarly, PV was calculated for the full task and its sub-phases by considering as an index the value of maximum muscle activation (Max) of the task, as shown in Table 1. In this case, the values assumed by PV are as follows: for the full task MaxPV = 25.39 [%], for bending MaxPV = 42.43 [%], for position holding MaxPV = 18.49, and for rising up MaxPV = −0.69 [%].

#### 4.1.2. IBA Analysis

With this methodology, it is possible to perform a muscular analysis that evaluates the assistance, resistance, and overall interaction with the exoskeleton. Considering the results in Table 1, the assistance provided by the exoskeleton is always much greater than the resistance. In the full task, Figure 1b, the AII is 22.44%, the RII is 1.22%, and the OII is 21.23%. During bending, position holding, and rising up, Figure 1d–e, the indices are proportional to the muscle energetic contribution required to perform the specific sub-phase of the full task (e.g., the amount of energy considered in the bending is different from position holding). As shown in Figure 1a, bending is the phase requiring the highest back muscle activation. From the values in Table 1, it is also the phase in which the exoskeleton provides its maximum support. The AII is 30.99%, the RII is 1.87%, and the overall interaction OII results in an assistance of 29.12%. Position holding represents a static phase with very low muscle activation. The analysis here highlights the slight differences in the muscle activities reported for this task. The results are AII = 18.99%, RII = 0.50%, and OII = 18.49%. To conclude, the rising up phase is when the exoskeleton contribution is lowest. The AII is 10.72%, the RII is 0.39%, and the overall contribution is an assistance of OII = 10.33%.

### 4.2. Discussion

To verify the validity of the proposed method, we report a mathematical demonstration. Considering the sampling rate as a constant and replacing $\Delta_{t}$ with *N* (total number of samples), it is possible to simplify Equation ((Equation 2)). Then, (Equation 1) can be derived from (Equation 5) by substituting VAL with the expression of the mean EMG calculated for the Exo and NoExo modality ($Mean\text{\_}EMG=\sum_{i=1}^{N}\left[EMG\right]/N$), as follows: (6)$$\begin{array}{c} OII=AII-RII=\frac{\sum_{i=1}^{N}\left[\hat{EMG_{NoExo}\left(i\right)}-\hat{EMG_{Exo}\left(i\right)}\right]\delta_{t}-\left|\left[\overset{ˇ}{EMG_{NoExo}\left(i\right)}-\overset{ˇ}{EMG_{Exo}\left(i\right)}\right]\right|\delta_{t}}{\sum_{i=1}^{N}EMG_{NoExo}\left(i\right)*N*\delta_{t}}=\\ =\frac{\sum_{i=1}^{N}[\hat{EMG_{NoExo}\left(i\right)}-|\overset{ˇ}{EMG_{NoExo}\left(i\right)}|-\hat{EMG_{Exo}\left(i\right)}+\left|\overset{ˇ}{EMG_{Exo}\left(i\right)}|\right]}{I\text{\_}EMG_{NoExo}*N}=\\ =\frac{\hat{I\text{\_}EMG_{NoExo}}+\overset{ˇ}{I\text{\_}EMG_{NoExo}}-\left(\hat{I\text{\_}EMG_{Exo}}+\overset{ˇ}{I\text{\_}EMG_{Exo}}\right)}{I\text{\_}EMG_{NoExo}*N}=\frac{I\text{\_}EMG_{NoExo}-I\text{\_}EMG_{Exo}}{I\text{\_}EMG_{NoExo}*N}=\\ =Mean\text{\_}PV/100=\frac{1}{N}\frac{Mean_{NoExo}-Mean_{Exo}}{Mean_{NoExo}}=\frac{1}{N}\frac{\sum_{i=1}^{N}\left[EMG_{NoExo}\right]/N-\sum_{i=1}^{N}\left[EMG_{Exo}\right]/N}{\sum_{i=1}^{N}\left[EMG_{NoExo}\right]/N} \end{array}$$
where I_EMG is the numerical integration of EMG that equals the mean of the same signal multiplied by N, showing the physical consistency of (Equation 2) in the instantaneous evaluation of the effect of the exoskeleton on the muscle activity.

Recalling Section 2, it is worth mentioning that the probability density of the surface EMG signals is modeled as a band-limited Gaussian [26]. However, Figure 1f reports the probability density graphs of the EMG data of the full task and its sub-phases, and as can be observed, none of the distributions take the form of a Gaussian. In this case, metrics such as the mean value do not coincide with either the mode or the median and, therefore, it is neither the most recurrent value nor the value that divides the distribution equally as it would be with a Gaussian distribution [27], but its statistical significance varies according to the distribution shape.

Focusing on the results, when considering the evaluation of the maximum muscle activity peak (Max), due to the complexity of the task, the automatic selection of the peaks can be erroneous. Thus, the graphical interpretation and meaningful representation of the data to make a correct and truthful understanding become critical. Considering the full task MaxPV, in fact, there is an error that is not visible apart from carefully checking the trend signals in Figure 1a. MaxPV was calculated considering the signals’ absolute maximum in the NoExo and Exo configurations during the full task. However, the maximum activations of the two configurations occur at completely different phases of the task (bending with NoExo and rising up for the Exo), as shown in Figure 1a. Thus, MaxPV for the full task was calculated with a mismatch of temporal instants and, consequently, of sub-activities. Since the MaxPV index obtained is based on a purely numerical analysis that does not evaluate the trend of the muscle graphs over the activity cycle but only the distribution of the signal, the analysis suffers from a mismatch error, and the result lacks validity. In this case, a decision must be made whether to analyze the muscle peak corresponding to the hip flexion or extension phase and proceed consistently to calculate the PV. The analysis of the EMG MaxPV highlights the impossibility of generalizing to the entire task the results obtained by analyzing specific signal distribution values without considering the signals’ trend (see Section 2).

Focusing on the IBA methodology, it is crucial to consider and understand the significance of the analysis performed. Equation (Equation 2) determines the point difference between the two EMG signals and normalizes the result with the value of the integral associated with the NoExo signal. This value is proportional to the activation intensity of the motor units recruited to perform the task and to the duration of the task itself. The difference between the EMG signals, on the other hand, represents the percent deviation from the normalization value. If the difference is positive, muscle activity decreased, so there has been assistance; if it is negative, muscle activity increased, so there has been resistance.

As shown in the demonstration at the beginning of the section (Equation (Equation 6)), when the sampling frequency is constant, the OII and Mean_PV can be considered the same entity. Thus, the presented method and proposed indexes (AII, RII, and the OII) are reliable from both a mathematical and a meaningful point of view. Moreover, this method allows the overall interaction index (OII) to be separated into further indices (AII and RII) that evaluate the interaction in terms of assistance and resistance to movement. The AII and RII themselves are computed by considering instantaneous values of EMG signals. Therefore, they allow a timely assessment of the task by providing added value to what would be obtained by an assessment of average values not only in terms of assistance/resistance but also of instants in which these occur. Thus, as hypothesized in Section 2, compared with traditional methods, the IBA allows us to investigate several aspects of pHEI. Specifically, because of the structure of Equation (Equation 2), the muscle energy contribution and the temporal duration of the task (given by the use of iEMG to normalize the signal) are implicitly considered in the calculation of the I_Trend signal; the assistive and resistive contributions of the interaction are quantified as a function of the change in muscle activation through specific indices (AII and RII respectively in Table 1); the instants of the task when the changes in muscle activation patterns occur can be identified from Figure 1b–e. This also allows assessment of the correspondence between assistance/resistance and the task phase execution (e.g., during most of the bending task, the exoskeleton provides assistance to the worker, but at the end of the movement, when the worker is assuming a static posture, there is a slight resistance, Figure 1a,c).

Based on the analysis carried out on the data in Figure 1 and the results in Table 1, and considering the consistency linking the overall interaction indices (OII) with the mean percentage variation indices (MeanPV), and emphasizing the added value provided by the IBA analysis in terms of task assistance and resistance quantification and their temporal identification, we believe the method is suitable to be used in an analysis that can be expanded to a wider number of subjects and exoskeletons.

## 5. Exoskeleton Performances Comparison

Considering the IBA method to be valid, we apply this analysis to all the data collected in the experimental session described in Section 2. The objective is to compare the assistive performance of three different exoskeletons for ten workers. The analysis aims to investigate the effect of these exoskeletons by evaluating the differences in muscle activations not only for spinal erectors but also for hip extensors and flexors. The following are the static and dynamic task results and discussions.

### 5.1. Results

#### 5.1.1. Static

Figure 4 shows the signals processed to perform the static task analysis. Figure 4a–d shows the erector spinae activity, Figure 4e–h the biceps femoris, and Figure 4i–l the rectus femoris. The indices associated with each of these graphs are shown in Table 2.

Starting with the spinal erectors during the full task, Figure 4a, we can see very different behaviors. Laevo and StreamEXO have mainly positive trends. AII and RII are 18.16% and 2.22% for the former and 20.89% and 0.94% for the latter, respectively. XoTrunk, however, has a mostly negative trend confirmed by AII = 4.00% and RII = 13.75%. Thus, for the full task, the StreamEXO provides the highest overall value of assistance to the workers’ back with an OII = 19.95%. The Laevo generates very good support with OII = 15.94%, while the XoTrunk reports more resistance than assistance. In fact, its overall index is negative, OII = −9.75%. Delving more deeply into the analysis of the spinal erectors to evaluate the individual subtasks, as shown in Figure 4b–d, the results are similar. During the bending phase, the Laevo provides the maximum assistance with OII = 19.49%, the StreamEXO has slightly less assistance with OII = 17.59%, while the XoTrunk again has a high resistance underlined by the negative value OII = −22.01%. During position holding, maximum support is provided by the StreamEXO, OII = 37.21%, followed by the Laevo, OII = 34.38%, with the XoTrunk again having a negative value, OII = −8.96%. Finally, during rising up, the StreamEXO provides a total support OII = 12.41%, the XoTrunk has a positive overall assistance equal to OII = 6.36%, while the Laevo confirms that a passive system does not input new energy by showing an overall energy loss that results in a slight resistance given by OII = −0.32%.

For the biceps femoris, Figure 4e–h, where the effects of the exoskeletons are experienced on the hip extenders, the signals associated with the XoTrunk are globally above the other signals, meaning positive results. In the full task, Figure 4e, the Laevo and StreamEXO signals rarely have a negative value, and even then, this occurs for short stretches. The XoTrunk never goes below the 0. Overall, the XoTrunk provides the maximum support with OII = 37.80%, the StreamEXO is in the middle with OII = 23.91%, and the Laevo is the lowest at OII = 20.16%. The XoTrunk also has the highest value of assistance in the bending phase, with OII = 34.14%, while the other two exoskeletons have OII values below 13.00%. The overall indices during position holding are high and fairly uniform: the Laevo has OII = 41.35%, the XoTrunk OII = 48.97%, and the StreamEXO OII = 48.92%. Finally, during rising up, the assistive contribution differs greatly among the three exoskeletons. Laevo provides the lowest assistance, with OII = 9.29%, followed by StreamEXO, with OII = 18.91%, and XoTrunk provides the highest support, with OII = 31.88%.

The hip flexors are analyzed by measuring the activation of the rectus femoris, as shown in Figure 4i–l. The signals associated with the Laevo are generally towards the top of the graphs, while those associated with the XoTrunk are more at the bottom. To reinforce this, the RII indices associated with the Laevo and StreamEXO have values close to 0, while RIIs associated with XoTrunk take values between 3.60% and 7.87%. Overall, the Laevo achieves the highest assistance values, with OIIs always greater than 16%, except during rising up, where OII = 13.91%. The StreamEXO maintains a positive trend with values ranging from OII = 4.43% during rising up to OII = 10.84% during bending. The XoTrunk, on the other hand, always reports more resistance than assistance. Its OII values are always negative and range from OII = −0.50% during bending to OII = −7.86% during position holding.

#### 5.1.2. Dynamic

The results for the signals during the dynamic task are reported in Figure 5. The lifting and lowering activities included in the dynamic task are analyzed, taking into account the effect of exoskeletons on the back muscles, as shown in Figure 5a,d, the hip extensors, as shown in Figure 5b,e, and the hip flexors, as shown in Figure 5c,f. The indices associated with these graphs are shown in Table 3.

Considering the spinal erectors in Figure 5a,d, the highest values are associated with the use of the StreamEXO, while the lowest values are with the XoTrunk. The indices in Table 3 confirm the results. During lifting, the assistive indices associated with all three exoskeletons are notably greater than the resistive indices. Overall, Laevo reports OII = 10.14%, XoTrunk OII = 6.58%, and StreamEXO OII = 20.99%. During lowering, the Laevo and the StreamEXO report low resistive indices, while the XoTrunk reports RII = 12.33%. Overall, the maximum assistance is reported by the StreamEXO with OII = 20.20%. The Laevo reports an assistance OII = 8.81%, while the XoTrunk has more resistance than assistance with OII = −5.17%.

For the biceps femoris shown in Figure 5b,e, the signal trends overlap frequently, but the XoTrunk seems to reach the highest assistance values. During lifting, the maximum resistance is given by the Laevo with RII = 5.80%, while the XoTrunk reports RII = 0.00%, and for the StreamEXO RII = 2.30%. Overall, the XoTrunk is the exoskeleton that reports the maximum assistive contribution for the hip extensor with OII = 27.90%, followed by the StreamEXO with OII = 10.77%, and finally, the Laevo reports more resistance than assistance, OII = −3.23%. During lowering, the Laevo and StreamEXO increase their negative contribution, with RII = 8.75% and RII = 14.22%, respectively. Thus, the XoTrunk reports the maximum assistance also for the lowering with OII = 24.10%, followed by the Laevo with OII = 3.07%, and last is the StreamEXO with OII = 2.03%.

For the rectus femoris, Figure 5c,f, the Laevo is the exoskeleton with the greatest assistive contribution. During lifting, the Laevo reports AII = 22.78%, for the XoTrunk AII = 0.65%, and for the StreamEXO AII = 9.64%. The XoTrunk is the only exoskeleton with a resistive index that deviates from 0; its index is RII = 21.01%. This overall contribution shows high assistance from the Laevo, OII = 22.77%, medium assistance from the StreamEXO, OII = 8.98%, and a markedly resistive effect from the XoTrunk, OII = −20.36%. During lowering, the Laevo once again has the greatest assistance with AII = 19.69%; the XoTrunk, in contrast, has the largest resistance with RII = 28.63%, while the StreamEXO balances the two contributions. Considering overall indices, the Laevo reports high assistance with OII = 19.69%, the XoTrunk has high resistance with OII = −27.21%, and the StreamEXO has only a slight resistive contribution, OII = −1.55%.

### 5.2. Discussions

The results above show the validity of the proposed analysis in both qualitative and quantitative terms, respectively, with the graphs and indices. From the figures, it is possible to understand in which phases of the task the signals are positive/negative and, thus, the specific instants during which the exoskeleton requires the user to generate less or more physical effort. Then, by calculating the indices associated with these trends, it is possible to quantify in percentage values the assistive/resistive contributions of the exoskeleton and evaluate its effectiveness on muscle activity reduction/increase. This knowledge can effectively aid the enhancement of control strategy design by guiding the developer to modify the strategy to improve the pHEI during the phases of the task characterized by resistive muscle behaviors.

#### 5.2.1. Static

During the evaluation of static activity, considering the analysis performed on the spinal erectors, the exoskeleton reporting the highest assistive contribution is the StreamEXO. In both the full task and its subphases, except bending, the StreamEXO reports the highest OII values. This result proves that sector/task-specific exoskeleton development can considerably impact muscle reduction, maximizing the benefits. During bending, on the other hand, the StreamEXO reports an OII = 17.59% while the Laevo OII = 19.49%. This slightly negative result of Stream may be due to the system not responding as quickly as a fully passive one. Regardless of these small numerical differences, the outcome is that the StreamEXO adequately assists all phases of the task. The Laevo also assists suitably in all tasks except rising up, where the overall interaction index is close to 0. The XoTrunk, on the other hand, shows a completely different behavior, producing more resistance than assistance in all phases of the task except rising up, where OII = 6.36%.

Considering the biceps femoris, all exoskeletons contributed positively to muscle support. Looking at the RII columns associated with this muscle, the resistive values are always close to zero, and the maximum deviation is below 2.50%. Considering the OII, the exoskeleton bringing the highest muscle support to the hip extensor is the XoTrunk, whose OIIs are never below 30.00%. Next is the StreamEXO, whose OIIs range from 9.59% during bending to 48.92% during position holding, and finally, there is the Laevo, whose OII values range from 9.29% during rising up to 41.35% during position holding.

Finally, considering the assistance provided to the rectus femoris, each exoskeleton responds very differently. Specifically, the Laevo reports the minimum value of overall assistance during the rising up phase, OII = 13.91%, while in all other phases and in the full task, the value is always higher than 16%. The StreamEXO, on the other hand, reports more dissimilar values ranging from OII = 4.43% during rising up to OII = 10.84% during Bending. Finally, the XoTrunk consistently reports more resistance than assistance, with overall indices ranging from OII = −0.50% during bending to OII = −7.86% during position holding.

In general, the StreamEXO provides the most support in the static task, regardless of the muscle group under analysis. Thus, its design and control strategy is suitable for the application and could generate potential impact in real applications, but the contribution provided during rising up could be improved. The Laevo follows with slightly lower OII values, a remarkable result considering that being a passive exoskeleton, it has a fixed control strategy defined only by hip flexion angle. Lastly, the XoTrunk reports excellent support values for the biceps femoris but in all other muscle groups, it has more negative than positive contributions. The control strategy adopted with this system performs poorly for the activity under analysis.

Looking more closely at the position holding task, reported in Section 3.2, this is of particular interest, as it is not clear if StreamEXO or Laevo is performing better. The former, in fact, reports slightly higher assistive values for the back erectors and hip extensors muscles, but the latter reports notably better support on the hip flexors muscles. Still, considering the difference in weight between the two exoskeletons, the control strategy adopted in the StreamEXO means it can compete with a device that weighs less than half its weight. Considering XoTrunk, we can state that a non-sector/task-specific exoskeleton design, unsuitable software, higher weight, and an ergonomic structure with a margin of improvement can dramatically penalize the device’s performance.

#### 5.2.2. Dynamic

In the dynamic task, the contribution of each exoskeleton changes significantly according to the muscles under analysis. For spinal erectors, the maximum assistance is provided by the StreamEXO, whose OIIs are always above 20%. Second is the Laevo, with OII values around 10%, while the XoTrunk, during the Lifting phase reports, assistance with OII = 6.58%, but during lowering, reports a resistive contribution with OII = −5.17%. Once again this result suggests that a sector/task-specific exoskeleton and optimized software can improve the provided benefits.

For biceps femoris support, the XoTrunk provides the highest assistance, with OII values around 25%. The StreamEXO reports good support in the lifting phase with OII = 10.77% but is of little use for lowering OII = 2.03%. The Laevo, in this case, provides slight resistance during lifting and slight assistance during lowering, but in both cases, the values are around ±3%.

The rectus femoris, in conclusion, receives maximum support from Laevo, whose OII is around 20% for both phases. The StreamEXO supports lifting with OII = 8.98%, but lowering slightly hinders the user with OII = −1.55%. The XoTrunk always reports highly negative values, showing resistance greater than 20% during lifting and greater than 27% during lowering.

For the dynamic task, it is difficult to define the most suitable exoskeleton without considering the effects of each on the different muscle groups. From the point of view of lumbar support, the StreamEXO is the exoskeleton that reports the most advantageous indices, but its assistance decreases when analyzing the hip muscles, especially in the Lowering phase. The Laevo particularly effectively supports the rectus femoris and has positive indices even considering the spinal erectors, but loses effectiveness when analyzing the biceps femoris. The XoTrunk, finally, reports very good values of assistance to the biceps femoris, and very bad values on the rectus femoris, while for erector spinae, it assists in lifting but resists during lowering.

The results show that assisting dynamic tasks requires many more considerations than static ones. Assisting certain muscle groups while performing rapid or complex movements does not mean fully assisting the worker. Conversely, it could mean overstressing other muscles. This is the example reported by the OII indices associated with the XoTrunk when analyzing hip muscles, as shown in Table 3. Although the extension assistance is excellent, the flexion assistance is bad, so the total contribution cannot be considered positive. For the other two exoskeletons, however, the assistive contribution is significantly greater than the resistive one.

## 6. Conclusions

In this paper, we presented a method to evaluate by introducing indexes in the form of trends, the possibility of numerically quantifying and temporally identifying epochs of muscle activity reduction and increment due to the physical human–exoskeleton interaction. In addition, we applied this methodology to analyze the effect of three exoskeletons on user muscle activation. Although the devices tested are back-support exoskeletons, this work reports a comprehensive analysis to evaluate the effects induced not only on spinal erectors but also on hip flexors and extensors. The proposed integral-based assistance (IBA) methodology was validated and subsequently used to assess the performance of three exoskeletons while performing two tasks, one static and one dynamic, typical of the cable’s conducts renewal in the rail sector. Thanks to the IBA, we quantified the support provided to the different muscle groups and highlighted the task phases in which the exchange of energy was assistive (muscle activity reduced) or resistive (muscle activity increased). For instance, considering the muscle activation of the spinal erectors during static activity, maximum assistance is provided by the StreamEXO during position holding (AII = 37.21%), proving that the assistive strategy for static tasks performs well for this exoskeleton. On the other hand, maximum resistance comes from the XoTrunk during bending (RII = 24.60%), suggesting that the control strategy that assists back flexion for load release should be improved to aid the users instead of contrasting their movements. StreamEXO is the exoskeleton that assists users the most overall during the performance of static tasks by presenting high AIIs and reduced RIIs. Its support is overall synchronized with the user’s movement, as evidenced by the positive OII values reaching 37.21% and 48.92% for the back and hip extensors, respectively. Conversely, each exoskeleton overwhelmingly supports a different muscle group during the dynamic task, so the StreamEXO is particularly suitable for erector spinae support (OII = 20.99%), the XoTrunk for hip extensors (OII = 27.90%), and the Laevo for hip flexors (OII = 22.77%). The conjoint results of the static and dynamic tasks suggest that the development of sector-specific exoskeletons strongly improves the benefits provided to the user by reducing more muscle activity and that the IBA method is a useful tool for deepening human-exoskeleton physical interaction. In fact, this method provides indices that allow quantification of both the assistance and resistance received from the exoskeleton and the phases of the task where this occurs. Thus, the method can be used to optimize the control strategy implemented on the exoskeleton by carefully understanding which phases of the task require higher support or a better timing choice.

## Figures and Tables

**Figure 1 bioengineering-11-01231-f001:**
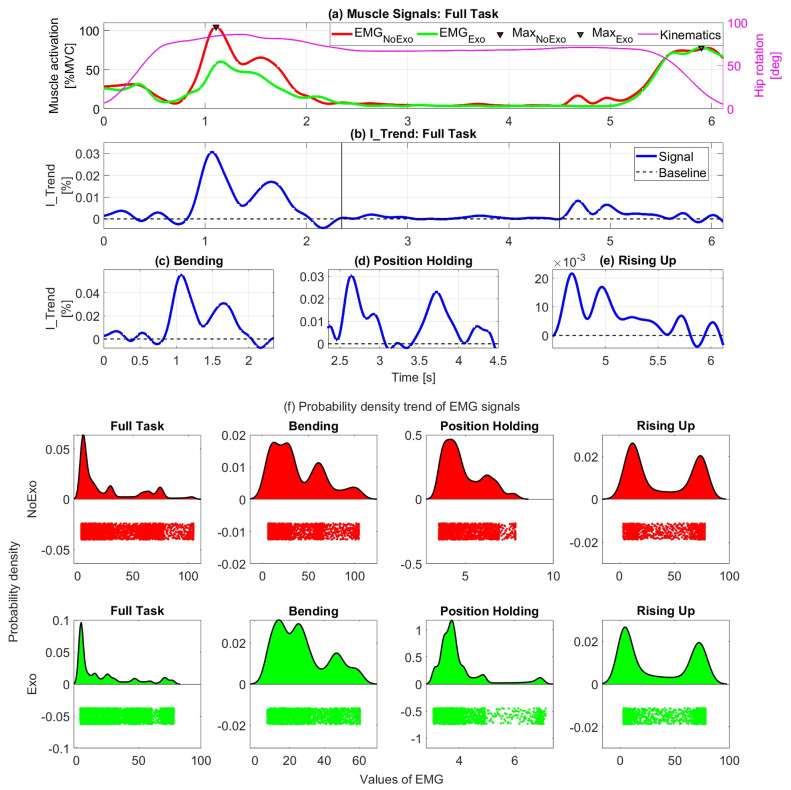
(**a**) EMG signals acquired with (Exo) and without (NoExo) an exoskeleton. I_Trend calculated for the: (**b**) full task, (**c**) bending and duct positioning, (**d**) bent position holding, and (**e**) rising up from the bent position. Figure (**a**) also shows the hip rotation signal, while figures (**b**–**e**) report the baseline to determine the assistance/resistance. The vertical black lines represent the sub-phase segmentation instants. Figure (**f**) shows the EMG signals’ probability density trends calculated for the NoExo (red) and Exo (green) configurations in all the task phases.

**Figure 2 bioengineering-11-01231-f002:**
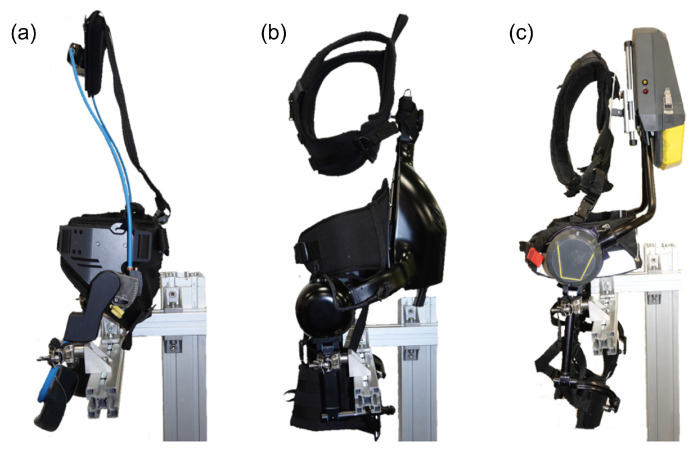
Side view of the exoskeletons tested with railway workers. (**a**) Laevo v2.56 (passive); (**b**) XoTrunk (active); (**c**) StreamEXO (active).

**Figure 3 bioengineering-11-01231-f003:**
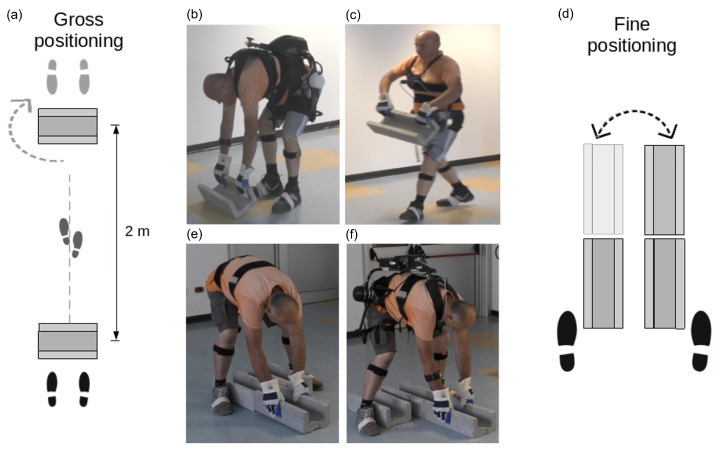
Experimental task explanation and examples. (**a**) Top view of the gross positioning setup; (**b**,**c**) worker during the lowering and carrying phases of the cable duct placement. (**d**) Top view of the fine positioning setup; (**e**,**f**) worker during the right and left positioning of the cable duct.

**Figure 4 bioengineering-11-01231-f004:**
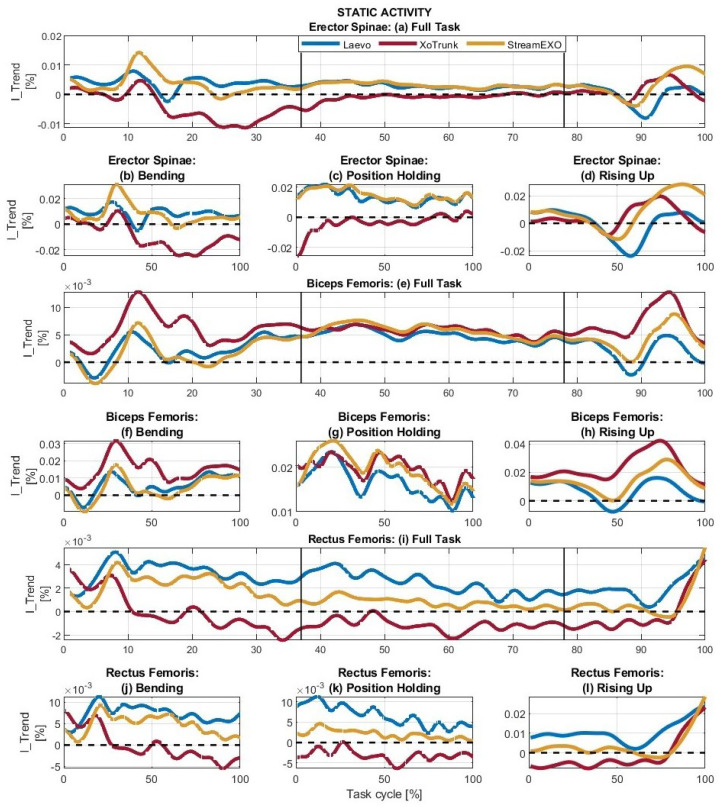
I_Trend calculated for the Full task and its sub-phases during the static activity. Each graph reports the signals associated with the Laevo, the XoTrunk, and the StreamEXO. (**a**–**d**) refer to the erector spinae muscles, (**e**–**h**) to the biceps femoris, and (**i**–**l**) to the rectus femoris. Vertical black lines represent the sub-phase segmentation instants.

**Figure 5 bioengineering-11-01231-f005:**
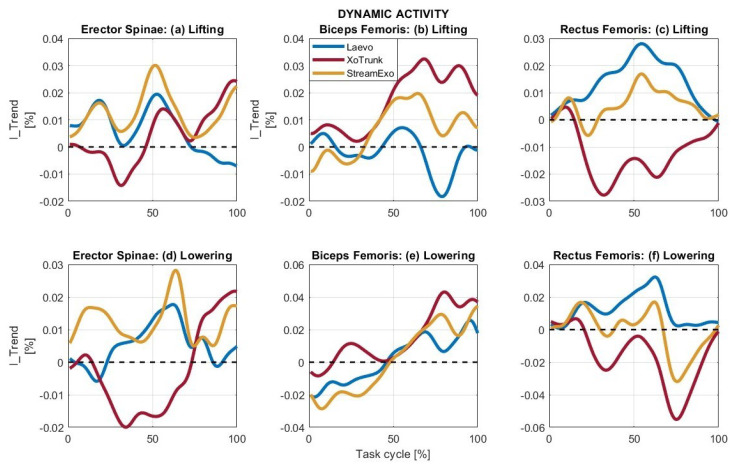
I_Trend calculated for the lifting and lowering phases of the dynamic activity. Each graph reports the signals associated with the Laevo, XoTrunk, and StreamEXO. (**a**,**d**) refer to the erector spinae muscles, (**b**,**e**) to the biceps femoris, and (**c**,**f**) to the rectus femoris.

**Table 1 bioengineering-11-01231-t001:** Upper: EMG average activation (Mean), maximum value (Max), and percentage variation (PV) calculated for the NoExo and Exo signals in Figure 1a on the full task and its sub-phases. Lower: assistive (AII), resistive (RII), and overall (OII) interaction indices calculated for the full task and its sub-phases (Figure 1b–e).

Standard Muscle Activity Indices												
	**Full Task**			**Bending**			**Position Holding**			**Rising Up**		
	**NoExo**	**Exo**	**PV**	**NoExo**	**Exo**	**PV**	**NoExo**	**Exo**	**PV**	**NoExo**	**Exo**	**PV**
**Mean**	26.63	20.98	21.23	38.25	27.11	29.12	4.89	3.99	18.49	38.64	34.64	10.33
**Max**	104.66	78.09	25.39	104.66	60.25	42.43	7.85	7.12	9.36	77.55	78.09	−0.69
**Integral Based Assistance Indices**												
	**Full Task**			**Bending**			**Position Holding**			**Rising Up**		
	**AII**	**RII**	**OII**	**AII**	**RII**	**OII**	**AII**	**RII**	**OII**	**AII**	**RII**	**OII**
**Interaction Indices**	22.44	1.22	21.23	30.99	1.87	29.12	18.99	0.50	18.49	10.72	0.39	10.33

**Table 2 bioengineering-11-01231-t002:** Assistive (AII), resistive (RII), and overall (OII) interaction indices calculated in the static activity for the full task and its sub-segmentations (Figure 4a–l). OII was calculated by subtracting RII values from AII. Thus, if OII is positive, the task is overall assisted, and if OII is negative, the task is overall resisted.

STATIC ACTIVITY									
**Integral Based Assistance: Erector Spinae**									
	**AII**			**RII**			**OII**		
	**Laevo**	**XoTrunk**	**StreamEXO**	**Laevo**	**XoTrunk**	**StreamEXO**	**Laevo**	**XoTrunk**	**StreamEXO**
**Full task**	18.16%	4.00%	20.89%	2.22%	13.75%	0.94%	15.94%	−9.75%	19.95%
**Bending**	19.96%	2.59%	17.97%	0.47%	24.60%	0.38%	19.49%	−22.01%	17.59%
**Position holding**	34.38%	0.90%	37.21%	0.00%	9.87%	0.00%	34.38%	−8.96%	37.21%
**Rising up**	5.64%	7.84%	14.69%	5.97%	1.48%	2.28%	−0.32%	6.36%	12.41%
**Integral Based Assistance: Biceps Femoris**									
	**AII**			**RII**			**OII**		
	**Laevo**	**XoTrunk**	**StreamEXO**	**Laevo**	**XoTrunk**	**StreamEXO**	**Laevo**	**XoTrunk**	**StreamEXO**
**Full task**	21.00%	37.80%	24.91%	0.84%	0.00%	1.00%	20.16%	37.80%	23.91%
**Bending**	14.12%	34.14%	12.04%	1.18%	0.00%	2.46%	12.94%	34.14%	9.59%
**Position holding**	41.35%	48.97%	48.92%	0.00%	0.00%	0.00%	41.35%	48.97%	48.92%
**Rising up**	10.50%	31.88%	18.91%	1.21%	0.00%	0.00%	9.29%	31.88%	18.91%
**Integral Based Assistance: Rectus Femoris**									
	**AII**			**RII**			**OII**		
	**Laevo**	**XoTrunk**	**StreamEXO**	**Laevo**	**XoTrunk**	**StreamEXO**	**Laevo**	**XoTrunk**	**StreamEXO**
**Full task**	16.07%	2.10%	7.65%	0.00%	5.60%	0.09%	16.07%	−3.50%	7.56%
**Bending**	16.61%	3.10%	10.84%	0.00%	3.60%	0.00%	16.61%	−0.50%	10.84%
**Position holding**	16.54%	0.01%	5.16%	0.00%	7.87%	0.00%	16.54%	−7.86%	5.16%
**Rising up**	13.91%	3.73%	4.90%	0.00%	5.98%	0.46%	13.91%	−2.25%	4.43%

**Table 3 bioengineering-11-01231-t003:** Assistive (AII), resistive (RII), and overall (OII) interaction indices calculated for the lifting and lowering phases of the dynamic activity (Figure 5a–f). OII is calculated by subtracting RII values from AII. Thus, if OII is positive, the task is overall assisted, and if OII is negative, the task is overall resisted.

DYNAMIC ACTIVITY									
**Integral Based Assistance: Erector Spinae**									
	**AII**			**RII**			**OII**		
	**Laevo**	**XoTrunk**	**StreamEXO**	**Laevo**	**XoTrunk**	**StreamEXO**	**Laevo**	**XoTrunk**	**StreamEXO**
**Lifting**	11.88%	10.32%	20.99%	1.74%	3.74%	0.00%	10.14%	6.58%	20.99%
**Lowering**	9.78%	7.16%	20.20%	0.97%	12.33%	0.00%	8.81%	−5.17%	20.20%
**Integral Based Assistance: Biceps Femoris**									
	**AII**			**RII**			**OII**		
	**Laevo**	**XoTrunk**	**StreamEXO**	**Laevo**	**XoTrunk**	**StreamEXO**	**Laevo**	**XoTrunk**	**StreamEXO**
**Lifting**	2.57%	27.90%	13.07%	5.80%	0.00%	2.30%	−3.23%	27.90%	10.77%
**Lowering**	11.82%	25.43%	16.25%	8.75%	1.33%	14.22%	3.07%	24.10%	2.03%
**Integral Based Assistance: Rectus Femoris**									
	**AII**			**RII**			**OII**		
	**Laevo**	**XoTrunk**	**StreamEXO**	**Laevo**	**XoTrunk**	**StreamEXO**	**Laevo**	**XoTrunk**	**StreamEXO**
**Lifting**	22.78%	0.65%	9.64%	0.01%	21.01%	0.66%	22.77%	−20.36%	8.98%
**Lowering**	19.69%	1.42%	7.48%	0.00%	28.63%	9.04%	19.69%	−27.21%	−1.55%

## Data Availability

The data presented in this study are available on request from the corresponding author.

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
