# Peer review of "Multi-Exoskeleton Performance Evaluation: Integrated Muscle Energy Indices to Determine the Quality and Quantity of Assistance"

_bioengineering, 2024, doi:10.3390/bioengineering11121231_

Round 1
Reviewer 1 Report
Comments and Suggestions for Authors
This paper presents a novel methodology for evaluating occupational exoskeletons by using integrated muscle energy indices, aimed at quantifying physical human-exoskeleton interaction (pHEI). Unlike traditional metrics based on peak or average muscle activation, this approach utilizes integrated electromyography (iEMG) to assess dynamic muscle engagement across task phases, revealing assistance and resistance patterns. It introduces indices like Assistive Interaction Index (AII), Resistive Interaction Index (RII), and Overall Interaction Index (OII) to analyze muscle energy expenditure. Through this method, the study compares three back-support exoskeletons, finding that task-specific exoskeletons significantly enhance user support during both static and dynamic tasks common in rail and construction work​​. This paper is well-structured and well-written. Before publication, there are some questions to be solved.
1. The author mentioned “Biomechanically robotic exoskeletons are structures that interact with the user’s musculoskeletal system to mutually transfer energy and power.”, more state-of-the-art can be cited: DOI: 10.34133/cbsystems.0115; DOI: 10.34133/cbsystems.0122; DOI: 10.34133/cbsystems.0141.
2. Could the authors provide more details on the iEMG preprocessing steps to ensure data consistency, especially for tasks with complex movements or time extensions?
3. How were synchronization and alignment between Exo and NoExo EMG signals managed across all task phases to prevent desynchronization effects on IBA indices?
4. What criteria determined the selection of specific muscles (back, hip flexors, and extensors) for evaluating the pHEI with exoskeletons? Were any other muscle groups considered?
5. What are the possible effects of physical variations (e.g., height, body mass) among participants on muscle activity reduction, and were any compensations applied in the analysis?
Reviewer 2 Report
Comments and Suggestions for Authors
In this study, the integrated muscle energy indices are presented to determine the quality and quantity of assistance for multi-exoskeletons performance evaluation.
Abstract is too long. The abstract should be cut short to highlight the key messages of the research. What are the limitations of the available methods that necessitated this proposed research?
In Introduction, the last paragraph, “In this paper, the proposed methodology is presented and validated….” The authors need to outline the issues associated with the existing indices that necessitated this research, What is the proposed methodology? What research gap the proposed paper intends to fill? What novel thing the paper is revealing?
In Methods, the texts for the description of Eqs should be put behind these Eqs. E.g.
Subsection 2.1 , try to put these Eqs (3)-(5) behind the description texts. While Figs should be put in front of the description texts, e.g. Fig.1, Fig2, …
Line 156, no reference number appears.
In discussions, What novel thing compared with the traditional indices is revealing in the paper?
In conclusion, the authors need to dwell more on the key results and their consequences because the values on AII=37.21%, RII=24.60%,….. are difficult to understand the meanings, it implies better/worse performance comparatively on the tested exoskeleton? Or something other meanings?
Round 2
Reviewer 1 Report
Comments and Suggestions for Authors
The authors have addressed almost all comments raised by reviewers. I would recommend the acceptance of this paper. Additionally, for comment 1, the cited references are suggested to added in the Introduction part, not line 22 in Abstract part.
Reviewer 2 Report
Comments and Suggestions for Authors
This paper can be accepted since the paper has been modified based on reviewer's comments.